# Antitumor Features of Vegetal Protein-Based Nanotherapeutics

**DOI:** 10.3390/pharmaceutics12010065

**Published:** 2020-01-15

**Authors:** Silvia Voci, Agnese Gagliardi, Massimo Fresta, Donato Cosco

**Affiliations:** 1Department of Health Sciences, University “Magna Græcia” of Catanzaro, Campus Universitario “S. Venuta”, Viale S. Venuta, I-88100 Catanzaro, Italy; silvia.voci@studenti.unicz.it (S.V.); gagliardi@unicz.it (A.G.); fresta@unicz.it (M.F.); 2Department of Experimental and Clinical Medicine, University “Magna Græcia” of Catanzaro, Campus Universitario “S. Venuta”, Viale S. Venuta, I-88100 Catanzaro, Italy

**Keywords:** antitumor compounds, gliadin, legumin, nanoparticles, soy protein isolate, zein

## Abstract

The introduction of nanotechnology into pharmaceutical application revolutionized the administration of antitumor drugs through the modulation of their accumulation in specific organs/body compartments, a decrease in their side-effects and their controlled release from innovative systems. The use of plant-derived proteins as innovative, safe and renewable raw materials to be used for the development of polymeric nanoparticles unlocked a new scenario in the drug delivery field. In particular, the reduced size of the colloidal systems combined with the peculiar properties of non-immunogenic polymers favored the characterization and evaluation of the pharmacological activity of the novel nanoformulations. The aim of this review is to describe the physico-chemical properties of nanoparticles composed of vegetal proteins used to retain and deliver anticancer drugs, together with the most important preparation methods and the pharmacological features of these potential nanomedicines.

## 1. Introduction

### 1.1. Nanotechnology and Cancer Therapy

Despite the significant advancements achieved in the field of antitumor therapy, the failure of treatment and mortality rates remain high, and cancer is classified as one of the main causes of death worldwide [1,2].

The advent of nanocarriers containing active compounds has favored the enhancement of the therapeutic index and the localization of anticancer drugs in tumor tissues, improving the efficacy of therapy and decreasing the drug resistance or the side effects that lead to treatment failure [3]. The resistance may be due to cellular mechanisms classified as changes in malignant cell biochemistry which include, but are not limited to, modified enzyme-specific activity, dysregulation in the apoptotic pathway and increased expression of efflux pumps like P-glycoprotein (P-gp) [4]. Among the various nanocarriers described in literature for antitumor application, polymeric nanoparticles were shown to be promising systems suitable for use in antitumor applications [5].

Polymeric nanoparticles can usually be classified as spheres or capsules as a function of their structure [6,7]. The preparation procedure and the materials used are able to modulate their morphology. Nanospheres are systems in which the drug is dispersed into the polymeric matrix or adsorbed onto its surface, while nanocapsules are “reservoir” systems in which the drug is dispersed or solubilized in an oily or aqueous core surrounded by a polymeric shell [8,9].

Polymeric nanoparticles are used to promote the solubilization of poorly-soluble anticancer drugs and modulate their pharmacokinetic features besides the distribution of entrapped compound(s) in body tissues, increasing their pharmacological efficacy in the neoplastic areas and minimizing side effects on healthy tissues [10,11]. Their small size provides several benefits such as a significant intracellular uptake [12] and the decrease of multidrug resistance (MDR). This is done by reducing the activity of the efflux pumps, as is true in the case of the P-glycoprotein (P-gp) [13]. 

During the preparation of a polymeric formulation, several variables should be considered in order to evaluate the compatibility of the biomaterial with the compound(s) to be entrapped and delivered, such as the structure of the system, the retention of the active compound and its release profile over time [14]. The scientific community is looking for biocompatible and biodegradable polymers developed from sustainable sources in order to come up with innovative drug delivery systems [15]. In this regard, plants are cost-effective, sustainable and renewable and can be considered a valuable source of natural polymers [16]. Plant-protein-based nanocarriers offer multiple benefits over lipid, carbohydrate and synthetic polymers and inorganic carriers (Figure 1) such as easy availability and low cost [17]. Plant-protein-based nanoparticles can interact by means of different mechanisms and chemical linkage with the bioactive compounds to be entrapped and with other derivatives used to modify their surfaces as a consequence of the functional groups present in their structures [18].

Vegetal proteins are characterized by greater amounts of hydrophobic amino acids as compared to animal proteins and are, therefore, generally more lipophilic [19,20]. Starting from this perspective, the aim of this review is to provide the state of the art concerning the development of drug delivery systems made up of the leading plant proteins currently being used in the pre-formulation investigations and that are available on the market, i.e., gliadin, legumin, soy and zein, besides their antitumor applications. Specifically, the chemical features of the described proteins, their influence on the preparation technique used to develop the nanosystems, and the physico-chemical properties of the resulting nanocarriers will be discussed. 

### 1.2. Mechanisms of Cell Uptake 

The active compound entrapped within a nanoformulation should be able to reach the intracellular environment in order to exert its therapeutic activity. Because of the hydrophobic character of the phospholipid bilayer, high polar and ionic systems are unable to cross the cell membrane by means of passive diffusion; therefore, mechanisms of active uptake represent the main phenomena that modulate the intracellular localization of nanovectors [21]. The uptake of polymeric nanoparticles is often characterized by an energy-dependent process known as endocytosis that enables them to reach the cytosol of cells and can be either phagocytosis or pinocytosis. The first one is a peculiar property of immune-cells, the main function of which is defense from external intruders such as viruses, bacteria and other pathogens; the second is employed by the cell to uptake small particles suspended in a fluid [22]. Pinocytosis is divided into clathrin- or caveolae-dependent endocytosis and clathrin- and caveolae-independent mechanisms [23].

#### 1.2.1. Clathrin-Dependent Endocytosis

This type of endocytosis is based on the interaction between the clathrin-coated pits, deriving from the invagination of the plasmatic membrane covered in a clathrin-1 grid, and the material that needs to enter the cell. The vesicles that detach from the membrane thanks to the GTPase dynamin are called clathrin-coated vesicles and are 100–150 nm in size; successively, clathrins are depolymerized and the fusion of the vesicular structures promotes the formation of endosomes fused to lysosomes [24]. It is important to note that a nanosystem is usually degraded by the enzymes and strong acid pH present when it reaches the lysosomes. This type of endocytic pathway is peculiar of nanoparticles made up of polylactide (PLA), poly (ethylene glycol-*co*-lactide) (PEG-*co*-PLA), poly-lactide-*co*-glycolide acid (PLGA), chitosan, and silica [25] and is inhibited by chlorpromazine, cytosol acidification and low concentrations of potassium [24].

#### 1.2.2. Caveolae-Mediated Endocytosis

The interaction between the membrane receptors and nanoparticles induces the formation of the so-called flask-shaped vesicles which derive from cholesterol and sphingolipid-rich domains of the plasmatic membrane known as lipid rafts. The vesicles are slowly internalized once they become pinched off from the membrane thanks to actin filaments, after which they reach the caveosomes, the equivalent of early endosomes but with a neutral pH [24]. Successively, the caveosomes reach the endoplasmic reticulum (RE) and can also favor the nuclear localization of the entrapped systems. This process needs more time with respect to the clathrin-dependent mechanism, involves smaller vesicles, and enables the cargo to escape enzymatic degradation [26].

#### 1.2.3. Caveolae Independent Endocytosis: Macropinocytosis

This pathway is known to be specific for cells lacking caveolae and clathrin and is exploited by folic acid, interleukin 2, growth factors, integrins, and epithelial cadherin [21,27]. Even though full comprehension of its mechanism is yet to be achieved, it has been reported that macropinocytosis modulates the uptake of lipids, proteins, receptors, viruses, bacteria, and toxins by means of vesicles known as macropinosomes, large vacuoles that can reach 10 μm in diameter, derived from wrinkles in the plasmatic membrane [28]. Macropinocytosis appears to be a mechanism used by positively-charged nanoparticles to reach the cytosol [29]. 

#### 1.2.4. Clathrin-Independent Endocytosis

The lack of specific markers and substrates makes clathrin-independent endocytosis a complex mechanism as compared to the aforementioned pathways [30] and can be dynamin-dependent or independent [31]. Flotilin and RhoA proteins are the mediators of the dynamin-dependent pathway, which respectively provide both pre-endocytic clustering and the uptake of the Epidermal Growth Receptor Factor (EGFR) or Interleukin-2 (IL-2). On the other hand, the dynamin-independent pathway which is mediated by Cdc42 and Arf6 provides the uptake of transmembranic protein CD44, toxins that bind glycolipids, receptors such as Glut 1 and Lat 1, and proteins of the major histocompatibility complex [32,33]. A potential mechanism for cargo internalization is related to either lipid or protein microdomains that can interact with the plasmatic membrane [34]. The development of specific inhibitors will be helpful for providing information on the clathrin-independent endocytosis mechanism in cell physiology, such as its involvement in the healing process of lesions of the plasmatic membrane [33,35].

## 2. Nanoparticle Features and Intracellular Localization

The sizes, shapes and surface charges are known to be parameters able to influence the cell uptake of nanoparticles because they can modulate their interaction with cells besides their passive/selective targeting [36]. Salatin et al., analyzed the effect that the shape of nanoparticles has on cell localization and demonstrated that elongated nanosystems can interact more efficiently with the cell membrane as compared to spherical ones, because the curvature of the particle surface is characterized by more accessible binding sites for the target cells. Moreover, the observation that rod-shaped nanoparticles were taken up much more efficiently by HeLa and Caco-2 cells was made [37]. Contrarily, Chitrani et al., reported that rod-shaped nanoparticles may have a larger area of interaction with the cell membrane than the spherical systems when the longitudinal axis of the rods interacts with the receptors, but this phenomenon favors a decrease in the amount of accessible receptors. This effect could also be attributed to the chemical entities present on the surface, such as surfactants, macromolecules or coating materials [38]. The same conclusion is reported in the work of Park et al. who demonstrated that rod-like nanoparticles had a reduced uptake in the ovarian cells of the Chinese hamster with respect to the spherically-shaped nanoparticles [39]. Another study demonstrated that smaller rod-shaped nanoparticles reached the insides of cells quicker than larger nanostructures [40]. 

The influence of the size of nanoparticles on endocytosis has not yet been clarified. Some studies have observed that the mean diameter of nanoparticles significantly affects their intracellular uptake [41]. 

Desai et al., demonstrated that nanoparticles of 100 nm are more efficiently taken up than those with dimensions ranging between 0.5 and 10 µm when incubated with Caco2 cells [42]; 100 nm-PLGA nanoparticles showed more than two and six times greater uptake with respect to the systems characterized by a mean diameter of 1 and 10 μm, respectively [43]. Another important characteristic to be evaluated during the development of a nanovector is the zeta potential, which is a parameter of the surface charge of particles—the higher/lower its value, the greater the surface charge is. The zeta potential influences the stability of the particles. In fact, charged particles promote repulsion forces and avoid aggregation of the system, favoring a homogenous size distribution [44]. However, the surface charge of nanoparticles may also be “tuned” to modulate their circulation time; positively-charged nanosystems are characterized by a higher rate of non-specific localization and a shorter plasmatic half-life as compared to those negatively and neutrally charged [45,46].

The main variables that determine the amount of adsorbed proteins, such as opsonin, are the surface characteristics of nanoparticles, such as surface charge and hydrophilicity/hydrophobicity. In general, it is important for a nanocarrier to have a neutral, hydrophilic surface in order to contrast the adsorption of plasmatic proteins so it can escape recruitment by the macrophages [47]. Xiao et al. demonstrated that negatively-charged nanoparticles are less up-taken by the liver and, consequently, can more efficiently deliver the entrapped active compounds to the tumor cells [48]. On the other hand, negatively-charged spherical particles can potentially bind to available cationic sites on the macrophage surface and be recognized by the scavenger receptors [36]. Moreover, it was recently demonstrated that cationic nanoparticles induce the creation of nanoscale cavities in the phospholipid bilayers that promote their cell uptake, bypassing the endocytic pathway [49].

## 3. In Vivo Fate of Nanoparticles

An exhaustive comprehension of the in vivo fate of nanoparticles is yet to be reached [50]. Following intravenous injection, the first entities encountered by the nanoparticles are plasmatic components, complement elements, and immunoglobulins. The phenomenon of opsonization aims to remove the particles by coating their surfaces with opsonins that drive them to the endothelial reticulum system (RES) [51,52]. As previously reported, this particle–protein interaction mainly depends on the surface characteristics of the carriers and it is crucial for the systemic distribution of the nanosystems [53,54]. In fact, nanoparticles with mean sizes greater than 100 nm will be removed by the RES, while those having dimensions inferior to 10 nm will be excreted to a significant extent by the kidneys. A size range between 10 and 100 nm is considered suitable for a nanosystem to be able to escape the RES and avoid renal clearance [55]. The microenvironment of solid tumors is a specific compartment characterized by fragile fenestrated blood vessels [56] with an absent basal membrane or else characterized by a peculiar thickness [57] and high interstitial pressure levels [58]. The immune and inflammatory cells, fibroblasts, lymphocytes, and signaling molecules can modulate the localization of the nanocarriers into the neoplastic area [59]. The peculiar features of these tissues promote the high permeability of the blood vessels and the accumulation of the colloidal systems in the tumor by means of the so-called enhanced permeability and retention (EPR) effect [58,59] (Figure 2). 

Cancer treatment can follow two main targeting strategies, i.e., passive and active approaches [60]. The first represents the ability of particles characterized by suitable sizes to accumulate in the tumor tissues by way of the previously described EPR effect. This phenomenon is a non-selective process that occurs thanks to the aforementioned, peculiar morphology of the neo-vessels and endothelial tissues of the tumor combined with the reduced activity of the lymphatic system [61]. Active targeting is achieved by the linkage of specific molecules onto the surfaces of the nanosystems in order to promote their interaction with specific receptors that are over-expressed in the tumor tissue/cells (for example transferrin, folate, hormones, hyaluronic acid, etc.) [62,63].

## 4. Plant-Protein Based Drug Delivery Systems: Manufacturing Processes

Based on their structures, polymers can be classified as crosslinked, branched or linear derivatives, while they can be defined as elastomers, plastic or fibers as a function of their physical properties [64]. Based on their origin, they are classified as synthetic, semi-synthetic or natural [65]. Proteins, cellulose and starch or polysaccharides, such as alginates, are all natural polymers extracted from plants, microbes and animals [66,67]; while polyesters, polyanhydrides or polyamides are synthetic compounds that can be customized to provide the required mechanical and chemical properties for various applications [68]. Among natural polymers, proteins such as casein, albumin and silk are currently being used for anticancer purposes [69,70,71]. Plant proteins, such as zein and gliadins, have drawn great interest in the last decade because they have been used to develop various biocompatible nanocarriers for anticancer purposes [19]. 

Various techniques have been developed to obtain nanoparticles, enabling widespread modulation of their structure, composition and characteristics. The choice of the best preparation procedure should be performed as a function of the features of the polymer and the physico-chemical properties of the compound(s) to be encapsulated [72]. 

### 4.1. Coacervation Process

Drug dispersion of preformed polymers is a common technique used in the preparation of biodegradable nanoparticles and can be accomplished using coacervation, interfacial deposition or electrospray deposition methods [73]. Due to its simplicity and advantage of yielding colloidal systems, the phase separation or coacervation process is the method of choice for obtaining protein-based nanoparticles [74]. The principle of this method is based on the decrease of the solubility of the protein in an aqueous phase when a dissolving agent such as ethanol, acetone or others that favor a separation of phases is added. The addition of such substances induces conformational changes in the protein and reduces its solubility, thus allowing the formation of nanoprecipitates [75]. Specifically, when a stable size of the colloidal systems is reached, an increase in the particle number is obtained [76]. During the preparation of protein nanocarriers, protein molecules are subjected to conformational modifications based on their concentration, composition, crosslinking, and chemical changes, which are due to the ionic strength, pH and type of solvent used [77]. The molecular weight of proteins as well as their isoelectric point (IP) are other key factors to evaluate because the first modulates the drug entrapment efficiency, its release profile and the targeting features, while the latter influences the stability of nanoparticles in different environments because the systems start to aggregate at pHs that are close to the IP. In fact the IP can dramatically affect the residence time of the nanoparticles in the blood stream and consequently the release of the entrapped bioactive(s) [78]. It has been demonstrated that a basic pH promotes the formation of smaller colloids and high salt concentrations neutralize the charges of the particles, promoting protein precipitation as nanosystems [79]. The addition of surfactants is a widely-recognized approach for stabilizing poorly water-soluble biopolymers such as zein and gliadin [80].

### 4.2. Nanoprecipitation

Fessi et al. developed and patented the technique of nanoprecipitation for the preparation of nanoparticles suitable for drug delivery [81]. Nanoprecipitation is an easy technique for the preparation of nanoparticles, and it is useful for the encapsulation of both hydrophilic and hydrophobic drugs [82]. Its main advantage consists of the rapidity of the formation of the nanoparticles because it is a one-step procedure and a scale-up translation of this technique is available. This approach requires the association of two miscible solvents, resulting in the spontaneous production of the nanoparticles through phase separation. Ideally, one of the two solvents should be the only one in which the polymer and the drug can be dissolved. A modified technique of nanoprecipitation uses a co-solvent either to increase the drug entrapment effectiveness or to decrease the mean sizes of the nanosystems [83].

### 4.3. Antisolvent Precipitation

This technique is also known as phase separation, liquid-liquid dispersion, desolvation or solvent displacement and it requires that the bioactive compound be previously dissolved in an organic solvent and then added to the antisolvent phase with rapid mixing [84]. This approach induces a supersaturation of the continuous phase and generates a significant amount of nuclei; submicron particles can be obtained if the nucleating crystals can be arrested through stabilizers by means of steric or electrostatic mechanisms. Water is mostly used as an antisolvent for hydrophobic drugs. The solvent should be able to dissolve the drug in large amounts and with a fast diffusion rate in the antisolvent water, while the stabilizer requires a strong attraction for the droplets of the drug, a rapid diffusion rate and an efficient adsorption onto the colloidal surface at the water–solvent interface. Therefore, in order to produce submicron particles, a good solvent–stabilizer combination is essential. Low costs and easy up-scaling are the major advantages of this method [85].

### 4.4. Solvent Displacement

This technique is normally characterized by the addition of the aqueous phase (enriched with a stabilizer) to a second lipophilic phase containing the polymer and the drug. The polymer deposition at the interface between the fine oil droplets and the aqueous phase promotes the formation of nanocapsules preferably exploited for the encapsulation of hydrophobic compounds. Solvent displacement and interfacial deposition are comparable techniques, the main difference of which lies in the final product because interfacial deposition techniques favor the formation of nanocapsules while solvent displacement can produce both nanospheres and nanocapsules [12,86].

### 4.5. Electrospray

Electrospray is used to obtain nanostructures, fibers or particles, through deposition by means of an electrostatic force onto a conductive substratum. The obtained droplets are extremely small, about 10 nm, and the charge, size and movement of these droplets can even be modulated by an electric signal [87]. This method is commonly used for the development of gliadin and elastin-based nanoparticles [77]. The high voltage provided by an emitter is applied to the protein formulation in order to obtain a liquid jet stream through a nozzle that helps form an aerosolized-size liquid. The nanoparticles are generated and attracted by a counter electrode [87]. 

## 5. Zein-Based Antitumor Drug Delivery Systems

In order to be used for biomedical, pharmaceutical and alimentary applications, a biomaterial must have GRAS (Generally Recognized As Safe) classification, be biodegradable, economic, and amply available [15]. Zein, a protein found in the endosperm of corn kernels, fulfills all these requirements and is characterized by hydrophobicity, bio-adhesion and significant cell uptake features. For these reasons it can be used as a suitable material for obtaining biocompatible delivery systems [88]. The cytocompatibility of zein is influenced by the molecular orientation of the protein: In fact, the molecules of this natural polymer can interact with the surfaces of the biological substrates organizing into different structures as a function of the polar properties [89]. Zein can be divided into four groups based on solubility and sequence homology: α-zein (19 and 22 kDa), β-zein (14 kDa), γ-zein (16 and 27 kDa), and δ zein (10 kDa). Among these, α-zein represents 70–85% of the total amount of zein content, and γ-zein is the second most abundant derivative (10–20%) [90]. More than half of the aminoacids of zein are non-polar, e.g., leucine, proline, alanine, and phenylalanine, but the lack of lysine and tryptophan and the few residues of arginine and histidine contained in its structure favor the solubilization of this natural polymer in hydroalcoholic solutions [90]. Zein also possesses large amounts of glutamines that are able to confer a certain polarity to the structure. The amphiphilic properties of zein allow the aggregation of the protein into colloidal particles able to retain lipophilic and water-soluble compounds [91].

The protein can form ribbon-like structures due to the arrangement of the hydrophobic portions and the hydrophilic extremities that can give origin to nanoparticles formed by the repetition of spherical blocks of protein units of approximately 20 nm [92,93]. Several techniques have been used to develop zein-based nanoparticles including interfacial deposition, phase separation and the nanoprecipitation method. Zou and Gu encapsulated daidzin, an isoflavone contained in soybean leaves characterized by antitumor activity, in zein nanoparticles using a modified antisolvent method [94]. The emulsifier d-α-tocopherol polyethylene glycol 1000 succinate (TPGS), a FDA-approved food additive, was added to improve the stability of the formulation: In fact, zein’s isoelectric point is 6.2 and the lack of stabilizers promotes the aggregation of protein nanoparticles in aqueous solutions with neutral or physiological pHs. Contrarily, the addition of a stabilizer is able to modulate the colloidal surface properties; in the present case, the nanosystems were characterized by a mean diameter of ~200 nm and a slightly decreased zeta potential with respect to the surfactant-free zein nanoparticles, probably as a consequence of the insertion of the lipophilic residues of TPGS (the tocopherol portion) into the inner cores of the systems and the coating of their surfaces with polyethylene glycol molecules. TPGS favored the increase of the encapsulation efficiency of the drug up to 63% (0.1 mg/mL of bioactive initially used) and promoted a rapid cellular uptake within Caco-2 cells [95]. By means of transcytosis, part of the daidzin-loaded nanovectors were transferred to the basolateral side of the intestinal epithelium cells through the surface charge shift and the related proton sponge effect (from the anionic character at pH 7.4 to the cationic character at pH 4–5 of the endolysosomes). This precluded the precocious degradation of the nanosystems [94]. In fact, the presence of protonable amine residues in the polymeric structure induced a significant ion/water inflow followed by an increased osmotic pressure, swelling of the endosomal vesicles and their destabilization with the consequent release of the drug into the cytosol [96]. Moreover, TPGS was reported to enhance the accumulation of daidzin in the cells when contained in zein nanoparticles because it is a P-gp inhibitor. In vivo investigations performed using TPGS emulsified-zein particles showed an improvement of the oral bioavailability of daidzin after its nanoencapsulation within the nanosystems [97].

Podaralla and Perumal encapsulated the 6–7 dihydroxy coumarin in zein nanoparticles as a model of a lipophilic compound and obtained a drug release that lasted for up to 9 days [98]; the systems were obtained by means of a pH-related coacervation process and it proved essential to maintain the pH of the water phase above the isoelectric point of the protein in order to avoid particle aggregation. The addition of lecithin and Pluronic F68 as stabilizers prevented the formation of macrostructures because the two surfactants favored the decrease of the mean diameter (~365 nm) as compared to the surfactant-free formulation (~460 nm) or to the systems prepared using a single stabilizer (>500 nm) [98]. Even though a specific anticancer application was not addressed in the described investigation, coumarin and its derivates possess antitumor properties against various diseases, as was true in the case of the derivative RKS262, which showed significant pharmacological activity against ovarian cancer resistant to platinum-drug-based treatments as a consequence of the decreased Bcl-xl and Mcl-1 levels [99,100].

Successively, Karthikeyan et al., performed an in silico analysis of simulated zein interaction with aceclofenac, metformin and promethazine in order to predict the best compound for retention by the protein nanosystems [101]. The simulations were associated to in vitro experiments in order to verify the results that confirmed the preferential retention and controlled release of the hydrophobic drug aceclofenac as compared to the hydrophilic and amphiphilic drugs, due to its interaction with the residues of the lysine, asparagine, proline, leucine, and isoleucine present in the protein. The entrapment efficiency of aceclofenac was of 20–21% when the drug/protein ratio was 1:2 or 1:5, while the drug release from the zein matrix showed no burst effect and lasted 3 days [101]. 

The antitumor antibiotic doxorubicin is another active compound widely used in anticancer therapy [102]; its well-established antiproliferative activity is related to its property of DNA intercalation and inhibition of topoisomerase-II, free radical production and mediated damage to cell membranes, DNA and proteins [103]. The peculiar side effects of the drug (i.e., cardio- and myelotoxicity) promoted the development of innovative formulations able to efficiently deliver the bioactive into the tumor area as is true in the case of the PEGylated liposomal nanomedicine Doxil/Caelyx [104,105]. The nanoencapsulation of doxorubicin hydrochloride in zein nanoparticles was performed by Dong et al. through phase separation method, and sodium caseinate (CAS) was included as a stabilizer [106]. The entrapment efficiency of the colloids reached 90% (when 0.5 mg/mL of the drug were initially added), while the release profile was pH-dependent and characterized by a faster drug leakage at an acid pH with respect to that of the neutral pH; this is a remarkable feature because the formulation was designed to release the drug into the tumor microenvironment, characterized by an acid pH value, preventing a premature and unnecessary leakage of doxorubicin into healthy tissues. The mechanism of cell uptake was also investigated, and it was demonstrated that it was a macropinocytosis-related phenomenon [106].

The influence of CAS was investigated by Luo et al., who demonstrated a decrease of the mean particle sizes when the CAS:zein ratio was 2:1, and this also favored an enhanced intracellular uptake of the systems into CaCo-2 cells [107]. This effect was explained by the better interaction between the cell surface and zein nanoparticles promoted by the stabilizer; in fact, the uptake of this formulation was characterized by energy-dependent endocytosis, as was demonstrated by the inhibition of their intracellular localization, obtained when using colchicine and sodium azide [107]. 

The same preparation method, i.e., phase separation, was performed in order to encapsulate 5-Fluorouracil (5-FU) [108], a thymidylate synthase inhibitor [109,110], in zein nanoparticles for liver targeting [108]. The nanosystems obtained using a zein:5-FU ratio of 3:1 *w*/*w* were characterized by a mean diameter of ~115 nm and a negative Z-potential (−45 mV). The in vivo investigations, performed in order to evaluate the biodistribution of fluorescent zein nanoparticles in mice after IV injection showed a significant accumulation of nanoparticles in the liver after just 2 h (~56%) and the same trend was observed after 24 h, evidencing a liver uptake that was 2.79 times greater than when a solution of rhodamine was used [108]. The overall biodistribution of the nanosystems also demonstrated a significant localization in the spleen but this was due to the scavenger effect exerted by the RES organs [108].

Among the various surfactants available and normally used to develop polymeric nanoparticles, Gagliardi et al. recently demonstrated that 1.25% *w*/*v* of sodium deoxycholate (SD) can significantly improve the physico-chemical stability of yellow zein nanosystems, characterized by a mean diameter of ~100 nm, a low size distribution, and a negative Z potential at various pHs (4–10) [91]. The possibility of avoiding the use of organic solvents to solubilize paclitaxel (PTX), such as Cremophor EL, is an important target of pharmaceutical technology because these cause the most important side effects in patients treated with the conventional formulations [111]; therefore, albumin-based nanoparticles have been developed [112]. Other biomaterials are under investigation in order to modulate the biopharmaceutical properties of PTX after its encapsulation, and zein may represent an innovative therapeutic option to be used for the treatment of various tumors [113].

SD-stabilized zein nanoparticles were obtained by the nanoprecipitation method and were used to entrap PTX [113], a lipophilic antitumor compound that acts by binding the tubulin beta-subunit and avoiding the normal microtubular breakdown [114]. 0.3 mg/mL of the drug added during the preparation steps of zein nanoparticles were efficiently retained by the protein structure (40%), confirming the peculiar properties of the nanosystems to entrap hydrophobic drugs as a consequence of the favorable interaction that comes about between the active compounds and the biopolymer [113]. PTX was gradually released by the nanostructures (70% after 5 days) and its nanoencapsulation permitted an increase in its cytotoxicity on various human cancer cells, such as K562 and MCF-7, with respect to its free form solubilized in ethanol [113] (Figure 3). 

Several investigations carried out in the past 20 years have shown that curcumin exhibits antimutagenic activity and suppresses the formation of neoplastic lesions in many tumor models such as skin cancer [115].

In fact, curcumin is able to down-regulate NF-kB, AP-1, Erg-1, and MAPK, decreasing the expression of Ciclooxigenase-2 (COX-2), Lipoxygenase (LOX), Nitric Oxid Synthase (NOS), Matrix Metalloproteinases (MMPs), Tumor necrosis factor (TNF), Interleukin-1 (IL-1), and cyclin D1; besides modulating CYP450 activity and inhibiting the activation of oncogenes such as Ras, Fos, Jun, and Myc [116]. Despite these important features, a useful formulation of the polyphenol derivative is difficult to obtain as a consequence of its photo-instability, poor water solubility and easy degradation. The encapsulation of curcumin in zein nanoparticles was performed by Patel et al., who proposed the inclusion of the active compound within the protein matrix as a valid approach for bypassing the above-mentioned drawbacks of the drug [117]. The systems were obtained by the antisolvent precipitation method using various curcumin:zein ratios plus 2% *w*/*v* of CAS as stabilizer. Although polyphenols are renowned for their great affinity to proline-rich protein, a certain degree of precipitation of curcumin outside the polymeric matrix was observed when the curcumin:zein ratio exceeded 1:5 [117]. Considering the mucoadhesive properties of zein, the nanoparticles were tested using agarose gel and mucin in order to evaluate their properties of active interaction with the mucus layer, prolonging the residence time of the curcumin [117]; moreover, the ability to preserve the polyphenol from the gastrointestinal environment was analyzed for up to 3 h. The results confirmed the mucoadhesive properties of the zein nanostructures and their plausible application for the oral delivery of curcumin as a consequence of the protective features exerted by the polymeric matrix on the stability of the drug [117].

A specific approach for contrasting the drug resistance of tumors is the employment of a combination of two active compounds aiming at different targets in order to obtain a synergistic anticancer effect [118]. The use of biocompatible delivery systems able to retain two or more drugs in the same structure has been investigated over the last decade in order to exploit this feature and various formulations have been developed [119,120,121,122,123,124].

Zein nanoparticles were used to obtain a multidrug carrier containing exemestane (EXM), a third-generation aromatase inhibitor commonly used in the treatment of hormone-dependent breast cancer, and resveratrol (RES), a phytoalexin and polyphenol extracted from grapes [125]. The potential synergism would derive from the resveratrol’s inhibition of estrogen biosynthesis through suppression of aromatase activity and then antagonism towards estradiol-induced MCF-7 cell growth [125]. The two drugs were co-entrapped in oily-core zein nanocapsules prepared with/without glutaraldehyde (GLA) as a crosslinker and proposed for oral breast cancer therapy. The carriers were obtained using the polymer interfacial deposition technique, and the addition of GLA promoted a slight increase of the particle size, an enhanced drug encapsulation efficiency and a controlled release of the active compounds in simulated gastrointestinal fluids as compared to GLA-free formulations [125]. The authors demonstrated that the oily core prevents premature protein degradation and favors a slower leakage of the drugs after 24 h (~59% and ~31% of released EXM and RES, respectively). Moreover, the co-delivery of the two drugs in zein nanosystems demonstrated better antitumor efficacy against breast tumors thanks to the synergistic inhibition towards aromatase, reduced angiogenesis due to the decrease of VEGF (Vascular Endothelial Growth Factor) levels, and increased apoptosis by means of Caspase 3 activation [125].

Thapa et al. proposed another zein-based multidrug formulation containing a proteasomal inhibitor, Bortezomib (Bor), and a histone deacetylase inhibitor, Vorinostat (Vor), for the treatment of metastatic prostate cancer [126]. The rationale is based on the pharmacological effects exerted by the two active compounds, which promote the death of cancer cells through apoptosis [127]. The carriers were obtained following the phase separation method previously described, using lecithin and poloxamer 407 as stabilizers. The formulation prepared using 0.2 mg/mL of zein and 1.1 *w*/*v* and 1.2% *w*/*v* of poloxamer and lecithin, respectively, was characterized by a mean diameter of about 130 nm that increased to 150 nm when the two drugs were co-encapsulated in the same colloidal structure; moreover, the association of the active compounds promoted a decrease in the entrapment efficiency probably as a consequence of the saturation of the space available. The multidrug carrier was characterized by a drug release profile of 48 h that increased at acid pHs, representing a positive outcome for promoting the leakage of the active compounds into the acidic tumor masses. In detail, 30–35% of entrapped Vor was released from zein nanoparticles into an acidic buffer solution (ABS) while the drug leakage into a phosphate buffer solution (PBS) was ~15%. Likewise, ~22% of Bor was released into an acidic medium with respect to 10–12% of the drug in PBS. The in vitro tests performed on LNCaP, PC3 and DU145 cells confirmed the hypothesis of the synergistic effect the two drugs have, even though a decreased cytotoxicity was obtained on the LNCaP line due to the significant phosphoinositide 3-kinase/AKT survival signaling pathway, which is an apoptotic escape mechanism carried on by the androgen-receptor positive cell lines [126]. The Western blot analyses demonstrated the increased expression of p53, while the in vivo tests performed on PC3 tumor xenograft models confirmed the potential application of this nanomedicine for the treatment of human prostate cancer [126]. In detail, the BALB/c nude mice groups treated with Vor and Bor co-encapsulated in zein nanoparticles showed a ~7-fold reduction in tumor mass as compared to the control group [126].

## 6. Gliadin-Based Antitumor Drug Delivery Systems

The main proteins found in cereals are alcohol-soluble prolamins, such as maize zein, barley hordein and wheat gliadin [128]. Such proteins share some common features, as they are stored within protein bodies in the endosperm during seed maturation and can only be solubilized in 85% and 70% ethanol for zein and gliadin, respectively [129]. This peculiar characteristic may be correlated to the amount of amin oacids they contain: prolamins contain high concentrations of proline (up to 30%), glutamine (up to 40%) and other essential amino acids like aspartate and lysine in limited amounts [129]. Gluten proteins can be divided into four fractions based on their electrophoretic mobility: α-gliadin, with a molecular mass of approximately 25–35 kDa; β-gliadin 30–35 kDa; γ-gliadin 35–40 kDa; and ω-gliadin 55–70 kDa [130,131]. Their structure can be described as a hydrophilic core domain, abundant in glutamine and proline residues, surrounded by two external regions rich in hydrophobic amino acids. For gliadin aggregation, non-covalent bonds including hydrogen-, ionic- and hydrophobic bonds are essential. [132]. Gliadin is used for oral administration, because it is a material characterized by bioadhesive properties that favor its interaction via hydrogen bonding with the intestinal mucosa [133]; the lipophilic amino acids allow the arrangement of the protein in nanoparticles with mucoadhesive characteristics [134]. The amine and disulphide residues of gliadin are able to interact with mucin and are therefore commonly employed for the development of nanosystems proposed for the gastrointestinal delivery of drugs [135,136]. In the case of colon cancer, this type of bioadhesion can be used to promote the site-specific release of anticancer agents [137]. 

Sharma et al., developed gliadin nanoparticles containing PTX with the aim of obtaining a controlled release of the drug [138]. The nanocarriers were prepared by desolvation and film-hydration methods in order to select the best formulation to be used for the in vitro and in vivo investigations: the size, Z-potential and drug encapsulation efficiency showed that the second technique permitted the development of nanosystems with physicochemical properties suitable for systemic administration. The use of Pluronic 127 (P127) contributed to decrease the size of the nanoparticles as a consequence of a favorable interaction that occurs with gliadin, while its association to Tween 80 and Triton X-100 showed an increased mean diameter [138]. The stability conferred by P127 was confirmed by the evaluation of the particle size up to 110 days, showing a slight increase from ~160 nm to ~220 nm. The drug release profile, evaluated at 37 °C by the dialysis technique using 0.3% Tween80-PBS solution as receptor fluid, was characterized by a PTX leakage of 88% after 24 h following a Fickian model. The in vitro cytotoxicity of PTX as a free drug or encapsulated in gliadin nanoparticles, assessed on MCF-7 and MDA-MB-231 cells, showed an IC_50_ value of 6.3 μM in both cell lines for the colloidal formulation after 48 h, which was higher than that obtained using the free form of the active compound (IC_50_ 1.75 μM) [138]. Even though these results could be detrimental to justifying the nanoencapsulation of the active compound within protein nanoparticles, the real efficacy of this nanomedicine could be appreciated after its in vivo administration as a consequence of the favorable pharmacokinetic profile of the colloidal systems and the absence of organic solvents required to solubilize the lipophilic drug. 

Another lipophilic compound entrapped within gliadin nanoparticles was all-*trans*-retinoic acid (ATRA), a metabolite of vitamin A, characterized by antitumor activity [130]. ATRA is a valuable therapeutic choice in the treatment of many cancers because it promotes cell differentiation or apoptosis and is used in the treatment of acute promyelocytic leukemia and precancerous lesions such as actinic keratosis [2]. ATRA-loaded gliadin nanoparticles were prepared by dissolving the protein and the drug in a 7:3 ethanol:water solution using the desolvation technique; the resulting nanosystems were characterized by a mean diameter of ~500 nm, a quite neutral surface charge (~−3 mV), and a drug encapsulation of 76.4 μg per each mg of protein [130]. The influence of GLA as a crosslinker was also investigated, and no significant variations in terms of size, particle surface charge or stability were found; moreover, nanoencapsulation preserved the stability of ATRA from both thermal- and light-induced degradation, even though the nanosystems did evidence significant degradation after just 3 h when they were incubated with trypsin [130]. The in vitro release profile of ATRA from gliadin nanoparticles was characterized by a 20% drug leakage after the first 15 min of analysis, evidencing a peculiar burst phenomenon that was attributed to a certain amount of active compound adsorbed onto the surfaces or encapsulated in the peripheral sites of the polymeric carriers; contrarily, the second phase of drug release was more controlled and followed a zero-order release kinetic [130]. The use of GLA as a crosslinker can lead to undesired consequences towards the drug release profile of the systems, as well as towards the body of the host [139,140], so Joye et al., proposed an alternative for improving the time- and temperature-stability of gliadin-based nanoparticulates by a simple modulation of the pH [141]. In another experimental investigation, Gulfam et al. used gliadin nanoparticles to entrap cyclophosphamide in order to investigate the cytotoxicity of this formulation on breast cancer cells [142]. The nanoparticles were developed by means of electrospray deposition and compared to hybrid systems made up of gliadin and gelatin. The systems prepared with 7% of pure gliadin provided the best cytotoxicity profiles on the MCF-7 cells, evidencing a decrease of Bcl-2 levels and significant apoptosis of breast cancer cells. The release of cyclophosphamide reached 79% after 24 h and was characterized by a two-step mechanism that was similar to the data previously described by Ezpeleta et al. [142] (Figure 4). 

The potential application of curcumin as an anticancer compound was investigated by Sonekar and coworkers who encapsulated the drug into gliadin nanoparticles [143]. The desolvation method favored the development of nanoparticles with a mean diameter of about 200 nm and a negative Z potential (around −20 mV). Bare gliadin and gliadin conjugated with folate residues showed different release profiles; in fact, pure gliadin nanoparticles were characterized by a burst leakage of the drug with respect to the folate-systems, (33% and 10% of the initial curcumin initially entrapped was released, respectively), while after 12 h this trend inverted. Moreover, the role of folate was shown to increase the uptake of curcumin-loaded gliadin nanosystems in cells expressing folate receptors, an approach able to further improve the antitumor effect of the drug, even though curcumin already tends to accumulate preferentially in tumor cells [143,144]. 

## 7. Legumin-Based Anticancer Drug Delivery Systems

Legumin belongs to the so-called 11S globulin classes with sedimentation coefficients between 11S and 14S and is one of the main storage proteins found in pea seeds, characterized by an oligomeric structure with a six-subunit configuration. Legume-seed proteins are salt-soluble globulins and can be separated into two main fractions, i.e., legumin and vicilin, by repeated dilution and precipitation or heat treatment. The isoelectric point of legumin is 4.8 and 5.5 for vicilin; the amount of lysine residues is higher in vicilin while legumin is rich in arginine portions [145,146]. The size and shape of the subunit domains are similar in both proteins, while the molecular weight of legumin (360 kDa) is greater than that of vicilin (200 kDa) [95]. According to Argos et al., it is appropriate to classify legume storage proteins as being composed of three domains: one NH_2_-terminal (domain I, span 1), one core (domain II, span 2) and one COOH-terminal (domain III, span 4). The carboxy terminal is rich in hydrophobic residues in the form of beta-sheets and is highly conserved in both legumin and vicilin, while a combination of both helical and beta-sheet structures is retrieved in the core domain II [147]. Legumin was used for drug delivery application, especially by Mirshahi et al. with the aim of entrapping methylene blue, a phenothiazinium synthetic basic dye, as a model compound, by means of a pH coacervation method and the inclusion of GLA as crosslinker [148]. It is known that the crosslinking process involves interaction between the GLA and lysine residues, promoting a mean particle diameter of 220–250 nm. The encapsulation of the dye resulted in an increase in the mean particle diameter (~300 nm) and poor drug entrapment efficiency. Also in this case, a biphasic release profile of the active compound resulted due to the rapid leakage of the methylene blue adsorbed on the particle surfaces [148]. The results obtained by Mirshahi et al. are important for a plausible translation of the formulation into antitumor application; in fact, it was demonstrated that methylene blue can exert significant anticancer activity against non-small lung tumor cells by the inhibition of heat shock protein 70 [149]. One of the main advantages deriving from the use of vegetal protein-based nanoparticulates is related to the great biocompatibility of the material [150]; the immune-response elicited by legumin nanoparticles confirmed this peculiarity and in fact no specific modulation of the antibody titer by the protein nanosystems occurred following administration in rats. Various hypotheses have been proposed, including the role of GLA on the conformational modification of legumin, but this phenomenon is still under investigation [145].

## 8. Soy Protein-Based Anticancer Drug Delivery Systems

Soy proteins extracted from soybeans have been in use since 1959, thanks to their functional properties [151]; β-conglycinine and glycinine are the main storage proteins contained in soybeans, characterized by a sedimentation coefficient of 7S and 11S, respectively. The peculiar properties of soy, which is composed of polar, non-polar and charged amino acids such as glutamate, aspartate and leucine, allow the interaction with and retention of a variety of drugs [78,152]. Soy protein isolate (SPI) is obtained by extractions at various pH values and the addition of salt [153]. Teng et al. developed SPI nanoparticles containing curcumin by using the desolvation method and GLA as a crosslinker [154]. The amount of crosslinker suitable for interaction with two residues of lysine was calculated as 28 μg of GLA per each mg of SPI. The mean diameter of SPI nanoparticles containing curcumin was of ~250 nm and a slight augmentation was observed when the amount of loaded curcumin increased, while the surface charge was negative (around −35 mV) and was not influenced by the drug/protein ratio [154]. These nanoformulations showed different physico-chemical properties with respect to those of curcumin-loaded zein nanoparticles, which were characterized by a smaller size due to the presence of a stabilizer. An example is negatively-charged caseinate, able to provide strong electrostatic and/or steric interparticle repulsion, while avoiding the attractive van der Waals forces [155,156]. The release profile of curcumin from SPI nanoparticles was very similar to that of the gliadin- and legumin-based nanosystems previously described, confirming the biphasic leakage of the drug from the protein matrix as a function of time [154]. Successively, these nanovectors were conjugated to folic acid, showing peculiar targeting properties for human colon carcinoma cells [157]. The same approach was used by other research teams in order to deliver doxorubicin, increasing its cytotoxicity activity against SH-SY5Y, MCF-7 and 293 T cells [158].

Zhang et al., encapsulated docetaxel [159], an antineoplastic compound the activity of which is ascribed to cell-division suppression through the stabilization of the microtubules [160] within SPI nanosystems by means of the antisolvent precipitation and ultrasonication techniques [159]. Various soy protein isolate amounts were screened during preliminary testing, and it was observed that the nanoformulations prepared with 15 mg/mL of SPI at basic pH promoted a significant decrease in the mean diameter of the particles [159]. The encapsulation of the drug induced an increase in particle size, in a manner proportional to the amount of the drug initially added. The cellular uptake of SPI nanosystems containing docetaxel was exerted on A549 cells, demonstrating efficient tumor accumulation of the colloidal systems mediated by clathrin and lysosome endocytic pathways; the cytotoxicity exerted by docetaxel encapsulated within SPI nanoparticles was characterized by a lower IC_50_ and a greater apoptotic effect with respect to the free drug [159].

## 9. Conclusions

For several years, pharmaceutical research has been investigating the antitumor application of innovative nanodelivery systems with the aim of increasing the localization of the active antitumor compounds directly into specific body compartments, thus precluding side effects on healthy tissues [161]. Plant proteins have been shown to be potential biomaterials to be used in the development of novel nanomedicines because of their biodegradability, biocompatibility, low cost, and ample availability [162,163]. In this review, an overview of the most important applications of plant-based nanodelivery systems in anticancer therapy has been shown (Table 1). Much investigation and characterization of these systems needs to be done to exploit these formulations in pre-clinical and clinical practice; in fact, their in vivo fate, their interaction with plasmatic proteins following intravenous administration, and their stability and capacity to modulate drug leakage are all aspects that should be better evaluated in order to develop suitable, efficacious nanomedicines.

## Figures and Tables

**Figure 1 pharmaceutics-12-00065-f001:**
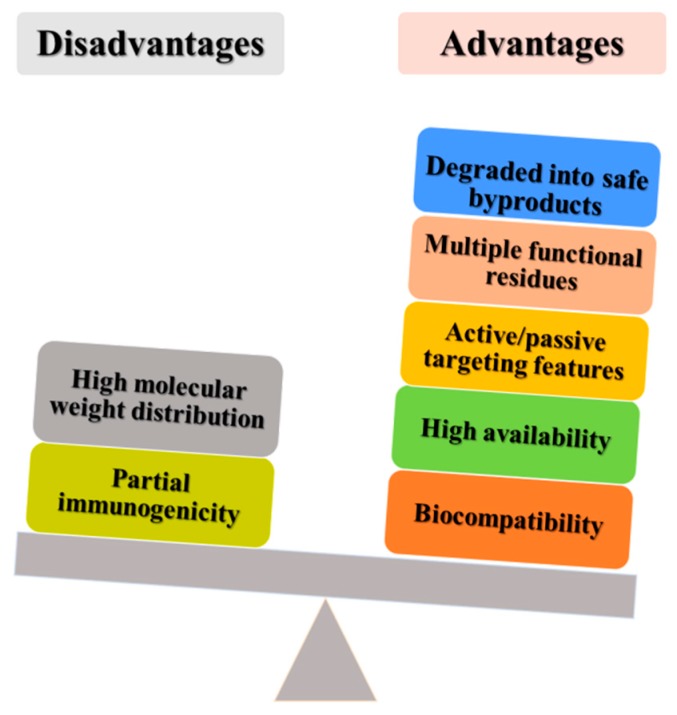
Main advantages and disadvantages of using plant proteins in drug delivery.

**Figure 2 pharmaceutics-12-00065-f002:**
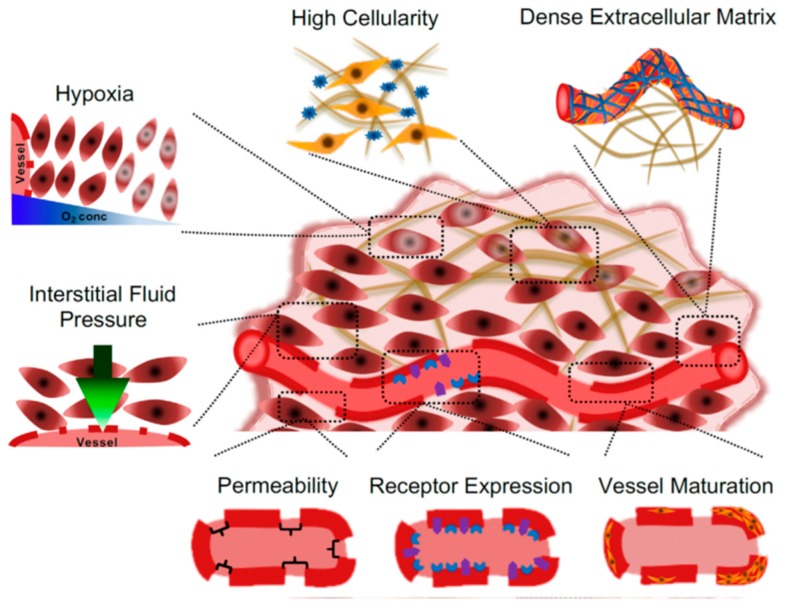
Schematic representation of the architecture of solid tumor vessels. Reproduced with permission from [59]. Copyright (2018) Elsevier.

**Figure 3 pharmaceutics-12-00065-f003:**
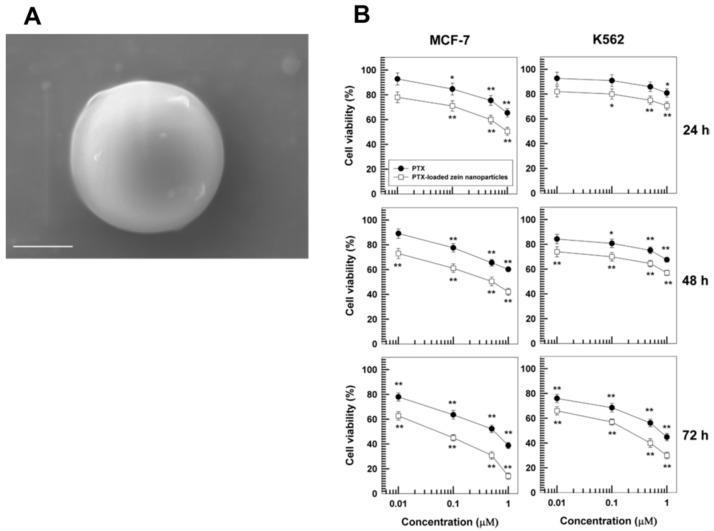
(**A**) SEM micrograph of sodium-deoxycholate stabilized zein nanoparticles (bar = 100 nm) and (**B**) in vitro antitumor activity of paclitaxel (PTX)-loaded nanosystems on various human cancer cell lines. * *p* < 0.05, ** *p* < 0.001 (with respect to the untreated cells). Adapted with permission from [91], copyright (2018) DOVE Medical Press and [113] copyright (2019) Elsevier.

**Figure 4 pharmaceutics-12-00065-f004:**
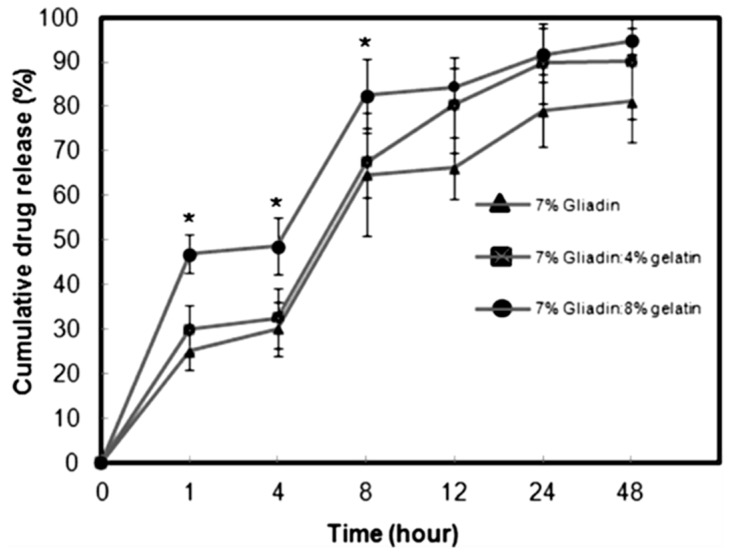
Release profile of cyclophosphamide from gliadin- or gliadin/gelatin-based nanoparticles as a function of the incubation time (* *p* < 0.05). Reproduced with permission from [142]; copyright (2012) ACS Publications.

**Table 1 pharmaceutics-12-00065-t001:** Composition, physico-chemical characteristics and antitumor application of vegetal protein-based nanoparticles.

Protein	Encapsulated Drug	Method of Fabrication	Mean Sizes (nm)	PDI	*Z*-Potential (mV)	Type of Treated Cancer Cells/Application	Reference
Zein	Daidzin	Modified Antisolvent Precipitation	150–200	0.2–0.3	20	/	[94]
	6,7-dyhydroxicoumarin	pH controlled nanoprecipitation	300–400	0.36	−11 ± 18	/	[98]
	Doxorubicin	Phase Separation	200–250	0.15–0.20	~−50	HeLa cells	[106]
	5-Fluorouracil	Phase Separation	100–150	/	−46 ± 1	Liver targeting	[108]
	Paclitaxel	Nanoprecipitation	<200	0.2	~−30	MCF-7; K562	[113]
	Curcumin	Antisolvent precipitation	109	0.12	~−30	Caco-2	[117]
	Exemestane and Resveratrol	Interfacial deposition	127 ± 3	0.13	~−32	Breast Cancer	[125]
	Vorinostat and Bortezomib	Phase Separation	150	0.20	−20/−30	Metastatic prostate cancer	[126]
Gliadin	All-*Trans* Retinoic Acid	Desolvation	~500	/	−4	/	[130]
	Paclitaxel	Desolvation and Film Hydration	160 ± 20	0.18	−21 ± 3	MCF-7; MDA-MB-231	[138]
	Cyclophosphamide	Electrospray deposition	~220	/	~18	Breast cancer cells	[142]
	Curcumin	Desolvation	~200	0.4	~−20	Colon cancer	[143]
Legumin	Methylene Blue	pH coacervation	250–300	/	~−40	/	[148]
Soy	Curcumin	Desolvation	220–286	/	~−35	/	[154]
Soy-Folate	Curcumin	Desolvation	170–300	/	−36	Caco 2	[157]
Soy-folate	Doxorubicin	Desolvation	232	/	−28, pH 5; −41, pH 10	293 T, MCF-7, SH-SY5Y cells	[158]
Soy	Doxorubicin	Desolvation	206	/	−20, pH 5; −30, pH 10	293 T, MCF-7, SH-SY5Y	[158]
Soy	Docetaxel	Antisolvent precipitation-ultrasonication	~250	0.3–0.4	~−10/−15	/	[159]

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
