# Peer review of "Antitumor Features of Vegetal Protein-Based Nanotherapeutics"

_pharmaceutics, 2020, doi:10.3390/pharmaceutics12010065_

Round 1

Reviewer 1 Report

The abstract seems in good shape.

Introduction is well written showing a good overview of the literature as it applies to this work. 

English:

The text should be revised by an expert in English grammar. 

Fig:

The figures are not self-explaining, the figure captions and/or the text referring to the figures should better explain the figure contents. Resolution of Figs. 3 and 4 are not good. Please improve the clarity of the scale mark and labels in your figures.

Reviewer 2 Report

Manuscript titled with "Antitumor features of vegetal protein-based nanotherapeutics" by Silvia Voci, Agnese Gagliardi, Massimo Fresta, Donato Cosco summarizes recent advancements in nanomedicines based on vegetal proteins. This review encompasses interesting progress in the field with adequate details especially for zein, gliadin, legumin, and soy-protein -based particles.

Author Response

The authors thank the Reviewer for the positive judgment.

Reviewer 3 Report

In the review entitled "Antitumor features of vegetal protein-based nanotherapeutics" authors bring to the reader the state of the art information  on the field in a well presented and easy to read manner. 

Manuscript is well structured but some minor points should be improved before publication. 

In line 78: start second sentence with "The uptake of polymeric nanoparticles..."

On item 1.2: there is no description of clathrin independent mechanism while other 3 mechanisms are detailed. 

On item 4.4: Authors are suggested to change the rorder of the paragraph by first presenting the technique charactherization (line 250) followed by the comparison (line 247) with interfacial deposition. 

In line 265: Define GRAS abbreviation

In line 300: give the approximate pH of endolysosomes reported in literature.

In line 361: what do authors mean by yellow zein?

Lines 377 to 382: Authors are suggested to change this paragraph concerning general PTX information to line 363, so that it comes before the description of a PTX formulation to improve its delivery. 

In line 364: check if it is the first time the word paclitaxel is shown and insert abbreviation (PTX)

In lines 370, 436, 502, 505 and 576: authors refer to " antitumor efficacy " when performing in vitro studies. Please change that to "cytotoxicity" and reserve antitumor efficacy for in vivo studies only. 

In line 403: This paragraph refers to nanocarriers co-encapsulating agents in synergistic ratios and two complete recent reviews on this theme are not cited.

In line 419: authors should bring quantitative information on the "slower leakage".

In line 432: authors should bring quantitative information on the "drug release profile" that is said to be increased.

In lines 438 and 587: The authors should provide more quantitative data regarding the increased in vivo efficacy of the formulations

In line 474: in the in vitro result reporting, authors mentioned two cell lines but only one set of IC50s

Concerning the figures presented along the text, is it all fine regarding copyright to reproduce them? 

Would it be possible to add figures of the basic structures of the zein, gliadin, legumin and SPI? That would significantly enhance the quality of the manuscript.

Author Response

The authors are very grateful to the Reviewer for his valued queries and advice. It is the opinion of the authors that the following changes in the manuscript have improved the quality of the paper. A response to each point raised in the main text has been shown in red.

Reviewer 2 (R2): In the review entitled "Antitumor features of vegetal protein-based nanotherapeutics" authors bring to the reader the state of the art information on the field in a well presented and easy to read manner. Manuscript is well structured but some minor points should be improved before publication.

In line 78: start second sentence with "The uptake of polymeric nanoparticles..."

Authors (A): According to the Reviewer’s suggestion the main text has been duly modified.

R2: On item 1.2: there is no description of clathrin independent mechanism while other 3 mechanisms are detailed.

A: In accordance with the Reviewer’s request, the clathrin-independent mechanism was discussed in the main text.

R2: On item 4.4: Authors are suggested to change the order of the paragraph by first presenting the technique charactherization (line 250) followed by the comparison (line 247) with interfacial deposition.

A: In response to the Reviewer’s suggestion, the paragraph was modified.

R2: In line 265: Define GRAS abbreviation

A: The GRAS abbreviation was properly defined as “Generally Recognized As Safe” in the main text.

R2: In line 300: give the approximate pH of endolysosomes reported in literature

A: According to the Reviewer’s comment, the information was added to the main text.

R2: In line 361: what do authors mean by yellow zein?

A: Commercial zein can be found as either a white or yellow powder. Yellow zein contains a high concentration (8–9%) of xanthophyll pigments, including lutein, zeaxanthin and cryptoxanthin. The hydrophobic xanthophylls are strongly bound to zein and various extraction methods have been used to decolorize the protein (Shukla, 2001, Ind Crops Prod.;13:171–92). The purity of yellow zein is 88–90%. In contrast to yellow zein, decolorized zein (white zein) contains very negligible amounts of xanthophylls (<0.001%), and the purity of white zein is >96% (Podaralla and Perumal, 2013, AAPS PharmSciTech, 13, 919-927).

R2: Lines 377 to 382: Authors are suggested to change this paragraph concerning general PTX information to line 363, so that it comes before the description of a PTX formulation to improve its delivery.

A: The main text was modified in accordance with the Reviewer’s suggestion.

R2: In line 364: check if it is the first time the word paclitaxel is shown and insert abbreviation (PTX)

A: The PTX abbreviation was properly inserted for the first time in line 381.

R2: In lines 370, 436, 502, 505 and 576: authors refer to " antitumor efficacy " when performing in vitro studies. Please change that to "cytotoxicity" and reserve antitumor efficacy for in vivo studies only.

A: In response to the Reviewer’s request, the term "antitumor efficacy" was replaced with "cytotoxicity" in the manuscript.

R2: In line 403: This paragraph refers to nanocarriers co-encapsulating agents in synergistic ratios and two complete recent reviews on this theme are not cited.

A: Recent references have been added to the main text, as suggested by the Reviewer.

R2: In line 419: authors should bring quantitative information on the "slower leakage".

A: This information has been duly added to the main text.

R2: In line 432: authors should bring quantitative information on the "drug release profile" that is said to be increased.

A: According to the Reviewer’s suggestion, the information was added to the main text.

R2: In lines 438 and 587: The authors should provide more quantitative data regarding the increased in vivo efficacy of the formulations

A: The information concerning the manuscript of Thapa et al., has been duly added to the main text. However, the data reported on line 587 is based on in vitro experiments and for this reason additional in vivo data cannot be provided.

R2: In line 474: in the in vitro result reporting, authors mentioned two cell lines but only one set of IC50s

A: In response to the Reviewer’s concern, the IC50 was duly described.

R2: Concerning the figures presented along the text, is it all fine regarding copyright to reproduce them?

A: In answer to the reviewer’s concern, the copyright of Figures 2, 3 and 4 has already been presented to the Editorial Office while Figure 1 was produced by the authors.

R2: Would it be possible to add figures of the basic structures of the zein, gliadin, legumin and SPI? That would significantly enhance the quality of the manuscript.

A: Useful information concerning the primary structures of the aforementioned proteins can be found at the following link: https://www.uniprot.org. This information was added to the main text.

Reviewer 4 Report

Minor points

-All references quoted in the text with the author’s first name must be homogeneous: 1st author et al.

-Page 9 Lines 385-386: many abbreviations need to be defined as COX, LOX, ...

Author Response

The authors are very grateful to the Reviewer for the valued queries and advice. It is the opinion of the authors that the following changes in the manuscript have improved the quality of the paper. A response to each point raised in the main text has been shown in red.

Reviewer 3 (R3): All references quoted in the text with the author’s first name must be homogeneous: 1st author et al.

Authors (A): According to the Reviewer’s suggestion, the references in the main text were duly revised.

R3: Page 9 Lines 385-386: many abbreviations need to be defined as COX, LOX, ...

A: In accordance with the Reviewer’s request, the various acronyms have been duly defined.